# Evaluation of Cardiovascular Toxicity of Folic Acid and 6S-5-Methyltetrahydrofolate-Calcium in Early Embryonic Development

**DOI:** 10.3390/cells11243946

**Published:** 2022-12-07

**Authors:** Zenglin Lian, Zhuanbin Wu, Rui Gu, Yurong Wang, Chenhua Wu, Zhengpei Cheng, Mingfang He, Yanli Wang, Yongzhi Cheng, Harvest F. Gu

**Affiliations:** 1Institute of Biological Chinese Medicine, Beijing Yichuang Institute of Biotechnology Industry, Beijing 100023, China; 2Shanghai Model Organisms Center, Inc., Shanghai 201203, China; 3School of Basic Medicine and Clinical Pharmacy, China Pharmaceutical University, Nanjing 210009, China; 4School of Life Science and Technology, China Pharmaceutical University, Nanjing 211198, China; 5College of Biotechnology and Pharmaceutical Engineering, Nanjing Tech University, Nanjing 211816, China; 6National Health Commission Key Laboratory of Birth Defect Prevention, Henan Institute of Reproductive Health Science and Technology, Zhengzhou 450002, China

**Keywords:** angiogenesis, cardiovascular toxicity, congenital heart disease, folic acid, 6S-5-methyltetrahydrofolate-calcium

## Abstract

Folic acid (FA) is a synthetic and highly stable version of folate, while 6S-5-methyltetrahydrofolate is the predominant form of dietary folate in circulation and is used as a crystalline form of calcium salt (MTHF-Ca). The current study aims to evaluate the toxicity and safety of FA and MTHF-Ca on embryonic development, with a focus on cardiovascular defects. We began to analyze the toxicity of FA and MTHF-Ca in zebrafish from four to seventy-two hours postfertilization and assessed the efficacy of FA and MTHF-Ca in a zebrafish angiogenesis model. We then analyzed the differently expressed genes in in vitro fertilized murine blastocysts cultured with FA and MTHF-Ca. By using gene-expression profiling, we identified a novel gene in mice that encodes an essential eukaryotic translation initiation factor (*Eif1ad7*). We further applied the morpholino-mediated gene-knockdown approach to explore whether the FA inhibition of this gene (*eif1axb* in zebrafish) caused cardiac development disorders, which we confirmed with qRT-PCR. We found that FA, but not MTHF-Ca, could inhibit angiogenesis in zebrafish and result in abnormal cardiovascular development, leading to embryonic death owing to the downregulation of *eif1axb*. MTHF-Ca, however, had no such cardiotoxicity, unlike FA. The current study thereby provides experimental evidence that FA, rather than MTHF-Ca, has cardiovascular toxicity in early embryonic development and suggests that excessive supplementation of FA in perinatal women may be related to the potential risk of cardiovascular disorders, such as congenital heart disease.

## 1. Introduction

Folate (also known as vitamin B9 and folic acid) is a water-soluble vitamin that is essential for DNA synthesis, RNA transcription, methionine synthesis from homocysteine, and other various chemical reactions involved in cellular metabolism [1,2]. Folate is especially important during periods of frequent cell division and growth, and adequate folate intake during pregnancy, lactation, and infancy is essential for maternal and infant health and normal growth [3,4]. Since vitamin B9 is an essential vitamin, the human body cannot synthesize it and requires its supplementation to maintain normal levels. Folic acid (FA) is a synthetic version of folate. It has been used as a substitute for folate in many countries around the world because it is more stable compared to folate [5]. However, FA is not generally found in natural food sources and cannot be directly absorbed and utilized by the human body until it is metabolized, while FA metabolism is achieved through the reduction of an enzyme called dihydrofolate reductase (DHFR) [6,7,8]. Evidence from rodent models indicates that high-dose FA supplementation may have deleterious developmental effects [9,10].

For the prevention of birth defects, including neural tube defects (NTDs), FA supplementation in pregnant women has been recognized and adopted as a global public health policy [11,12,13], but its role in congenital heart defects (CHDs) remains controversial [14,15]. The total incidence of CHDs in populations worldwide has increased [16]. A multicenter, double-blind randomized controlled trial failed to show any advantage of 4.0 mg FA versus 0.4 mg supplementation on the occurrence of congenital malformations [17]. Earlier studies suggested that FA has a protective role in decreasing the risk of preeclampsia [18,19], whereas recent studies have demonstrated that high-dose FA has no effect in preeclampsia [20]. Furthermore, a recent study reported that FA in the serum may mask the benefit of (6S)-L-5-tetrahydrofolate on renal outcomes. 6S-5-methyltetrahydrofolate is the predominant form of dietary folate in circulation and is used as a crystalline form of calcium salt (MTHF-Ca) [21]. Up until now, however, it has been unclear if there are any different effects between FA and MTHF-Ca on embryo development. Therefore, it is necessary to evaluate the toxicity and safety of FA and MTHF-Ca.

In the current study, we initially tested how FA induces the developmental toxicity, particularly in the embryonic cardiovascular system in zebrafish, and comparatively assessed the cardiotoxicity between FA and MTHF-Ca on zebrafish development. We then investigated the impact of FA and MTHF-Ca on the birth rate of newborn litters in C57BL/6J mice and examined clinical parameters related to heart function in newborn litters. We analyzed the differently expressed genes in in vitro fertilized murine blastocysts cultured with FA and MTHF-Ca because the activity of DHFR in rodents is high [8]. By using gene-expression profiling, we identified a novel gene in mice that encodes an essential eukaryotic translation initiation factor (*Eiflad7*), which is highly homologous with humans (*EIFIAX*) and zebrafish (*eif1axb*). Furthermore, we knocked down this gene in zebrafish by using a morpholino antisense to ascertain its role in cardiovascular morphogenesis and angiogenesis during early embryogenesis to explore the putative molecular mechanism. Finally, we found that the loss of *eif1axb* in zebrafish could cause global and pericardial edema, as well as vascular defects characterized by reduced vascularization in intersegmental vessels and decreased sprouting in subintestinal vein (SIV) vessels. Thereby, the findings from the current study will be helpful for better understanding the inconsistent findings among studies of FA on pregnancy outcomes and the toxicity and safety of FA and MTHF-Ca.

## 2. Methods

### 2.1. Toxicity and Safety Studies of FA and MTHF-Ca in Zebrafish

Adult zebrafish were maintained at 28.5 °C on a 14 h light/10 h dark cycle [22]. Five to six pairs of zebrafish were set up for nature mating every time. On average, 200–300 embryos were generated. Embryos were maintained at 28.5 °C in fish water (0.2% instant ocean salt in deionized water). The embryos were washed and staged as previously described [23]. The *Tg(fli1a:EGFP)^y1^* and *TG(zlyz:EGFP)* transgenic lines were established and characterized accordingly [24,25]. Zebrafish were purchased from SMOC (Shanghai Model Organisms Center, Inc., Shanghai, China) and fed with Artemia. The zebrafish facility at SMOC is accredited by the Association for Assessment and Accreditation of Laboratory Animal Care (AAALAC) International.

To determine the toxicity and safety of lead compounds, FA (Jiheng Pharmaceutical Co., Ltd., Hebei, China) and MTHF-Ca (Magnafolate pro, Jinkang Hexin Pharmaceutical Co., Ltd., Lianyungang city, China) zebrafish (*n* = 40, each group) were treated from 4 to 72 h postfertilization, and the mortality was recorded every 24 h. Dead zebrafish was defined as the absence of heartbeat under a dissecting stereomicroscope (Nikon SMZ745). In the initial tests, fourteen concentrations (0.1, 1, 2.5, 5, 10, 25, 50, 100, 250, and 500 μM; and 1, 2.5, 5, and 10 mM) were used for the compounds. Mortality curves were generated using GraphPad Prism 5.0 (GraphPad Software). The development phenotypes were also recorded.

### 2.2. Angiogenesis Studies in Zebrafish

To evaluate blood-vessel formation in zebrafish embryos, at 4 h postfertilization, *Tg(fli1a:EGFP)^y1^* zebrafish embryos were distributed into 6-well plates (thirty embryos per well) (BD Falcon) and treated with vehicle control (fish water), FA (2.5, 5, and 10 mM), or MTHF-Ca (2.5, 5, and 10 mM) for 68 h. All embryos were incubated at 28.5 °C. After treatment, embryos were anesthetized with 0.016% MS-222 (tricaine methane sulfonate, Sigma-Aldrich). Zebrafish were then oriented on their lateral side (anterior, left; posterior, right; dorsal, top), and mounted with 3% methylcellulose in a depression slide for observation by fluorescence microscopy. The mean length and diameter of intersegmental vessels (ISVs) and the area and length of SIVs were quantified with NIS-Elements D4.6 software (Nikon SMZ18).

In addition, fertilized one-cell *fli1a-EGFP* transgenic line embryos were injected with *eif1axb-MO* and control-MO. At 3 dpf, the embryos were dechorionated and anesthetized with 0.016% MS-222 (tricaine methane sulfonate, Sigma-Aldrich, St. Louis, MO, USA). Zebrafish were then oriented on their lateral side (anterior, left; posterior, right; dorsal, top), and mounted with 3% methylcellulose in a depression slide for observation by fluorescence microscopy. The phenotypes of complete ISVs (the number of ISVs that connect the DA to the DLAV) and SIVs were quantitatively analyzed.

### 2.3. Acridine Orange Staining for Apoptosis in Zebrafish

To assess the effects of FA or MTHF-Ca on heart apoptosis in zebrafish, at 4 h postfertilization, *Casper* zebrafish [26] embryos were distributed into 6-well plates (thirty embryos per well) (BD Falcon) and treated with vehicle control (fish water), folic acid (1, 2.5, and 5 mM), or MTHF-Ca (1, 2.5, and 5 mM) for 92 h. After treatment, embryos were immersed in 5 μg/mL AO (acridinium chloride hemi-zinc chloride, Sigma-Aldrich) in fish water for 1 h at 28.5 °C in the dark [27,28]. Next, zebrafish embryos were rinsed thoroughly in fish water three times (5 min/wash) and then oriented on their lateral side and mounted with methylcellulose in a depression slide for observation by fluorescence microscopy. 

### 2.4. TMRM Staining in Zebrafish

The mitochondrial membrane potential (ΔΨm) was estimated by monitoring fluorescence aggregates of TMRM (tetramethyl rhodamine methyl ester, perchlorate, and biotium) [29,30]. To determine the efficacy of TMRM staining in zebrafish embryos, at 4 h postfertilization, *Casper* zebrafish embryos were distributed into 6-well plates (thirty embryos per well) (BD Falcon) and treated with vehicle control (fish water), folic acid (1, 2.5, and 5 mM), or MTHF-Ca (1, 2.5, and 5 mM) for 92 h. After treatment, embryos were immersed in 1 μM TMRM in fish water for 1 h at 28.5 °C in the dark. Next, at 26 h postfertilization, zebrafish embryos were rinsed thoroughly in fish water three times (5 min/wash). Zebrafish embryos were then oriented on their lateral side and mounted with methylcellulose in a depression slide for observation by fluorescence microscopy.

### 2.5. Embryotoxicity Testing and Transcriptome Analysis in C57BL/6J Mice

To further evaluate the effects of FA and MTHF-Ca on cardiovascular development, we carried out experiments in the ovulated oocytes from female mice (C57BL/6J). Superovulation induced by exogenous gonadotropin treatment (PMSG/hCG) was applied to assess the quality of ovulated oocytes from female mice. In vitro fertilization procedure was then followed as previously described [31]. The fertilized eggs were randomly grouped for control with potassium simplex optimized medium (KSOM) (Elite-Media), KSOM-FA (60 ng/L), and KSOM-MTHF-Ca (60 ng/L). After culture for 24, 48, and 72 h, the blastocysts were collected, examined under a microscope (Olympus, Japan), and also stored in liquid nitrogen. The blastocysts of mice 72 h after in vitro fertilization were collected from three groups, i.e., control, FA, and MTHF-Ca. Total RNA was prepared by using RNeasy kit (Qiagen) according to the manufacturer’s instructions. An equal quantity of RNA from all samples was blended for cDNA library construction to obtain the transcriptome data. A normalized cDNA library was constructed with 1 μg of total RNA. The sequencing libraries were generated using NEBNext Ultra Directional RNA Library Prep Kit for Illumina (NEB) and sequenced on an Illumina HiSeq 2000 platform for pair-end reads. All sequences were processed and analyzed by Genewiz Bio-pharm Technology Corporation.

### 2.6. Zebrafish Microinjections

Gene Tools, LLC (http://www.gene-tools.com/, accessed on 1 December 2022) designed the morpholino (MO). Antisense MO (GeneTools) was microinjected into fertilized one-cell stage embryos according to standard protocols [32]. The sequences of the *eif1axb* translation-blocking and splice-blocking morpholinos were 5′-CTCCTTTTCCTTTGTTTTTCGGCAT-3′ (ATG-MO) and 5′-CAGCCTGAAGCTCTAAAATGCACCT-3′ (E3I3-MO), respectively. The sequence for the standard control morpholino was 5′-CCTCTTACCTCAGTTACAATTTATA-3′ (Gene Tools). The amount of the MOs used for injection was as follows: Control-MO and E3I3-MO, 2 ng per embryo; ATG-MO, 2 ng per embryo. Primers spanning *eif1axb* exon 2 (forward primer: 5‘-GCGACGTGGTAAGAACGAGAA-3′) and exon 5 (reverse primer: 5‘-GGCTTTCAGACTCCTAGCCT-3′) were used for RT-PCR analysis for confirmation of the efficacy of the E3I3-MO. The primer *ef1α* sequences used as the internal control were 5‘-GGAAATTCGAGACCAGCAAATAC-3′ (forward) and 5‘-GATACCAGCCTCAAACTCACC-3′ (reverse). The CDS region of *eif1axb* cDNA, including the *eif1axb*-ATG-MO target sequence, was cloned in frame into pcDNA3.1-EGFP for testing the effectiveness of *eif1axb* morpholino oligonucleotides (MOs).

### 2.7. Image Acquisition

Embryos and larvae were analyzed with a Nikon SMZ18 Fluorescence microscope and subsequently photographed with digital cameras. A subset of images was adjusted for levels, brightness, contrast, hue, and saturation with Adobe Photoshop 7.0 software (Adobe, San Jose, CA, USA) to optimally visualize the expression patterns. Quantitative image analyses were processed using image-based morphometric analysis (NIS-Elements D4.6) and ImageJ software (NIH, http://rsbweb.nih.gov/ij/, accessed on 1 December 2022). Inverted fluorescent images were used for processing. Positive signals were defined by particle number using ImageJ. Ten animals for each treatment were quantified and the total signal per animal was averaged.

### 2.8. RT-PCR

In zebrafish, total RNA was extracted from 30 to 50 embryos per group in trizol (Roche, Basel, Switzerland) according to the manufacturer’s instructions. RNA was reverse-transcribed using PrimeScript RT reagent kit with gDNA Eraser (Takara, Kusatsu, Japan). Quantification of the *eif1axb* gene expression was performed in triplicate using Bio-rad iQ SYBR Green Supermix (Bio-rad, Hercules, CA, USA) with detection on Real plex system (Eppendorf). Relative gene expression quantification was based on the comparative threshold cycle method (2^−ΔΔCt^) and normalized with the reference *ef1α* gene [33].

### 2.9. Bioinformatical Analysis

After total RNAs were constructed for the library, Illumina platform was used for high-throughput sequencing. Sequencing reads were assessed with Bcl2fastq (V2.17.1.14) to generate FASTQ files. Initial quality control of FASTQ files was performed with FastQC (V0.10.1) and Cutadapt (V1.9.1) to filter low-quality data, including contamination and redundant sequences introduced by adapters. Clean reads were compared with the mouse genome by Hisat2 (V2.0.1) and the gene expression of short reads based on fragments per kilobases per million reads (FPKM) was quantitated by Htseq (V0.6.1). The differentially expressed genes (DEG) were identified with DESeq2 (V1.6.3) in the bioconductor package. Volcano and dot maps were finished in R (V4.0.3) with the ggplot2 package. We obtained complete sequences of *Eif1ad7 (mice)* and *eif1axb* (zebrafish) from GenBank and obtained similar human *EIFIAX* sequences through the use of the online basic local alignment search tool (BLAST) of NCBI. Orthologs of *Gm5662* among human, mouse, and zebrafish sequences was obtained by alignment of DNAMAN (V6.0.3). Based upon RNA-seq expression data, gene-set variation analysis (GSVA) (v1.45.2) was conducted to predict the specific pathways.

### 2.10. Statistical Analysis

All data are presented as mean ± SEM. Statistical analysis and graphical representation of the data were performed using GraphPad Prism 5.0 (GraphPad Software, San Diego, CA, USA). Statistical significance was performed using a student’s *t*-test, ANOVA, or χ^2^ test, as appropriate. In RNA-seq and the related functional enrichment analyses, the false-discovery rate (FDR) and q value were utilized as FDR-based measures of significance for differential expression analysis.

## 3. Results

### 3.1. FA, but Not MTHF-Ca, Has Cardiovascular Toxicity In Vivo

Chemically, the molecular weights of FA and MTHF-Ca are 443.40 and 497.52, respectively. The structural formulae of these two compounds are represented in Figure 1A,B. We started to evaluate the potential toxicity and safety of FA and MTHF-Ca in a zebrafish model system. NaHCO_3_ was used as a solvent because FA is insoluble in water. The test experiments with NaHCO_3_ were initially conducted and the data showed that the morphology of zebrafish embryos in the control group (NaHCO_3_ 20 mM) was similar to what was observed in untreated eggs. Formal experiments for the evaluation of toxicity and safety of FA and MTHF-Ca in zebrafish were subsequently performed. The endpoints of the FA and MTHF-Ca concentrations were established from 0.1 to 10,000 μM of FA and MTHF-Ca. We found that FA, but not MTHF-Ca, caused cardiotoxicity, which resulted in developmental defects in zebrafish embryos. Cardiotoxicity was observed in dose ranges from 250 to 10,000 μM of FA. Cardiotoxicity was observed in 8 of 40 exposed zebrafish embryos at 250 μM of FA, while exposure at the highest concentration (2500 μM of FA) had a 100% cardiotoxicity in all 40 exposed embryos. The mortality rates at 5000 and 10,000 μM of FA were 2/40 and 32/40 (Figure 1C,E), respectively, and 100% cardiotoxicity in the surviving embryos. However, there was no cardiotoxicity or embryonic defects in MTHF-Ca-exposed embryos (Figure 1D,F). Further analyses indicated that the pericardial area and SV-BA distance in FA-treated embryos were increased compared to the embryos exposed to MTHF-Ca (Figure 1O,P), while heart rate was lower than what was observed in MTHF-Ca (Figure 1Q). Thus, the effects of FA on sinus venosus and bulbus arteriosus (SV-BA) distance and heart rate exist in a dose-dependent manner in zebrafish. The represented images are shown in Appendix A.

### 3.2. FA Is Different from MTHF-Ca and Leads to Abnormal Vascular Development

To determine whether the appearance of zebrafish heart, heart rate, and blood circulation are related to blood vessels, we used a zebrafish angiogenesis model to assess the efficacy of FA and MTHF-Ca. As shown in Figure 2A–H, FA, but not MTHF-Ca, impaired the formation of zebrafish SIV vessels in a dose-dependent manner. In control embryos, SIVs developed as a smooth, basket-like structure over the yolk at 3 dpf (Figure 2A,E, yellow arrow). In contrast, embryos treated with FA presented with fewer ectopic SIV segments (Figure 2B,C, asterisks). Quantification of the area and length of SIVs showed a significant decrease in FA-treated embryos (Figure 2I–J). In the images of trunk regions at 3 dpf, with the vascular structures being visualized by eGFP fluorescence and labeled ISV and DLAV (dorsal longitudinal anastomotic vessel), it showed regular development in the control-treated embryos. Compared with control or MTHF-Ca-treated groups, the FA-treated embryos had fewer incomplete and thinner ISVs (Figure 2N, yellow arrows). In control embryos, the parachordal vessels (PAVs) formed normally (Figure 2K, red arrows). Compared with control or MTHF-Ca, FA prevented the formation of the parachordal vessels (PAVs), the precursor to the lymphatic system (Figure 2N). Quantification of the mean length or diameter of ISVs showed a significant decrease in FA-treated embryos. (Figure 2S,T) Compared with control- or MTHF-Ca-treated groups, FA-treated embryos presented a lower number of incomplete and thinner ISVs. In control embryos, PAVs formed normally. Compared with control or MTHF-Ca, FA prevented PAV formation, the precursor to the lymphatic system. Therefore, FA, but not MTHF-Ca, causes vascular defects, secondary to impairing the formation of the zebrafish SIVs and ISVs.

### 3.3. Neither FA Nor MTHF-Ca Induce Zebrafish Heart-Specific Apoptosis and Macrophage Migration

As described above, FA had a pronounced cardiotoxic effect in a dose-dependent manner. As such, it was necessary to analyze the possible mechanism underlying the observed heart apoptosis. A special zebrafish strain known as Casper was treated with either FA (1, 2.5, and 5 mM, Figure 3D–L) or MTHF-Ca (1, 2.5, and 5 mM, Figure 3M–U). We found that both FA- and MTHF-Ca-treated embryos exhibited few or no apoptotic cells in the heart, (Figure 3V) even if FA induced cardiotoxicity. The results indicated that cardiotoxicity of FA was unrelated to excessive apoptosis. Neither FA nor MTHF-Ca induced heart-specific apoptosis in zebrafish. Furthermore, macrophage migration was examined to determine whether FA-induced cardiotoxicity was caused by inflammation. Compared with the control group, both FA and MTHF-Ca treatments showed the normal distribution of labeled cells and no induced macrophage migration was observed in the heart (Figure 3 A_0_–U_0_). Thus, the cardiotoxicity of zebrafish caused by FA treatment is not related to inflammatory lesions (Appendix A).

### 3.4. Identification of an Essential Eukaryotic Translation Initiation Factor

DHFR is an important enzyme that is essential for the conversion of FA to active folate. Bailey and Ayling previously reported that the activity of DHFR in rats is extremely high compared with human DHFR activity [8]. In the current study, we detected DHFR in the liver of mice and found its level was high, as expected (Appendix A). Because of the higher activity of DHFR in rodents, we carried out experiments in mouse blastocysts cultured with FA and MTHF-Ca (Appendix A). By using RNA-seq transcriptomic analyses, we detected the DEGs in mouse blastocysts cultured with FA and MTHF-Ca. Of these DEGs, the expression of *Gm5662 was* most significantly different between FA- and MTHF-Ca-treated blastocysts (Appendix A). *Gm5662* is an mRNA sequence recorded in the mouse genome database, but the gene had not yet been identified. Taking advantage of comparative gene mapping, we performed a blast search for a homology analysis between Gm5662 (*M. musculus*) and AAP36772.1 (*H. sapiens*). Human *EIF1AX,* encoding an essential eukaryotic translation initiation factor, was found to have homologous identities with Gm5662 in amino acid sequences by 92%. Therefore, *Gm5662* was identified as the *Eif1ad7* gene in mice (*M. musculus*) and *eif1axb* in zebrafish (*D. rerio*). The gene is conserved in Eukaroyota and shares high homology among humans (*EIF1AX*), mice (*Eif1ad7*), and zebrafish (*eif1axb*) (Figure 4A). Further experiments with qRT-PCR indicated that the *eif1axb* gene was downregulated by FA, but not MTHF-Ca, in zebrafish at 12 to 48 h postfertilization (Figure 4B).

### 3.5. Loss of eif1axb Phenocopies FA-Induced Cardiovascular Defects

To explore whether FA inhibition of *eif1axb* causes cardiac developmental disorders, we further analyzed the phenotypes after *eif1axb* knockdown using MO antisense. The effectiveness of *eif1axb* knockdown is shown in Appendix A. The knockdown efficiency of *eif1axb* was confirmed by RT-PCR and qRT-PCR. Compared with control zebrafish (Figure 5A), *eif1axb* knockdown caused pericardial oedema, reduced contractile force, and reduced precardiac blood congestion as signs of manifest heart failure (Figure 5B,C). Embryonic defects increased, while survival rates decreased (Figure 5D,E). Quantification of the pericardial area shows a significant increase in *eif1axb* morphants (Figure 5F). Images of trunk regions taken at 3 dpf, with the vascular structures visualized by eGFP fluorescence and labeled ISV and DLAV, showed regular development in the embryo injected with the control MO. Compared with control MO, embryos injected with *eif1axb*-MO present thinner ISVs (Figure 5K,L, yellow arrows). In control embryos, the parachordal vessels (PAVs) formed normally (Figure 5J, red arrows). Compared with control, MO knockdown *eif1axb* prevented PAV formation, the precursor to the lymphatic system. The boxed regions are shown at higher magnification in the bottom panels (Figure 5N,O). Quantification of the mean length and diameter of ISVs showed a significant decrease in *eif1axb* morphants (Figure 5P,Q). Therefore, downregulation of the *eif1axb* gene inhibits early vascular development in zebrafish.

## 4. Discussion

FA supplementation is primarily used in pregnant women to prevent congenital malformations, such as NTD, and indeed it has achieved remarkable results [11,12,13]. In contrast to the adverse effects of folate deficiency on embryonic development, the possible negative effects of FA appear negligible [34,35,36]. It is generally believed that the safety margin for FA administration is quite high, while high-dose FA supplementation has already appeared in use. FA supplementation is crucial during pregnancy and the previous reports on the prevention of CHD are conflicting [37,38,39]. It is unknown whether FA has cardiovascular toxicity in early embryonic development. By using a zebrafish animal model, in the current study, we have demonstrated that FA inhibits the angiogenesis, interferes with normal development, and causes early cardiovascular and embryonic developmental defects. There are data suggesting that excessive antiangiogenesis early on in the pregnancy may play a role in the origin of some CHDs, which are associated hypertensive disorders, particularly in women with preterm preeclampsia [40,41]. Interestingly, the previous studies have shown that longer FA supplementation appears to increase the risk of pregnancy-associated hypertension and preeclampsia [19,42,43]. The data from our study are in agreement with the above-mentioned studies. In general speaking, the related persistence of a high-resistance maternal uterine circulation and maternal antiangiogenic stress response are all considered to be involved in the pathogenesis of preeclampsia. A schematic illustration to demonstrate that excessive FA treatment has cardiovascular toxicity and leads to angiogenesis inhibition in zebrafish is shown in Figure 6.

A previous study demonstrated that the 20 mg/kg FA-supplemented diet has effects on heart development (increased ventricular septal defects) [10]. DHFR is hundreds of times more active in the liver of rodents than in humans (Appendix A) [8]. Therefore, it is hard to use rodents for in vivo studies to analyze the effects of FA on cardiovascular and embryonic development. We further performed RNA transcriptome analysis and found that Gm5662 in the mouse genome was differentially expressed in FA cultures compared to controls. By using the concepts and methods of comparative genome analysis, we found that Gm5662 is highly homologous to the human eukaryotic translation initiation factor 1A (*EIF1A*) gene, thereby confirming that Gm5662 is a mouse *Eif1ad7* gene. Due to the high homology, the gene is conserved across species, suggesting that it may be an ancient gene that plays a key regulatory role in embryonic development. So far, information about this gene is very limited, and the downstream pathways regulated by this gene remain unclear. However, a recent study showed that *EIF1A* is a novel component of the Ago2-centered RNA-induced silencing complex and enhances Ago2-dependent RNAi and miRNA biogenesis [44]. We suggest that the inhibitory effect of FA on cardiovascular and embryonic development was mainly in the early embryonic stage. This is supported by the qRT-PCR and average litter size results. This gene was found to be downregulated early under FA treatment. Thereby, the gene dysfunction would lead to early embryonic lethality rather than specific congenital malformations.

In zebrafish, the sequence of Gm5662 is closer to *eif1axb*. We employed a zebrafish model and MO knockdown technology to characterize the biological function of the zebrafish *eif1axb* gene, which is orthologous to the human *EIF1AX* gene, during early development in zebrafish. The zebrafish *eif1axb* gene is 81.5% homologous to that of the human *EIF1AX* gene and is 99.3% conserved at the protein level. Knockdown of gene function by MO is the most familiar genetic approach to study genes of interest in zebrafish [45]. Loss of *eif1axb* in zebrafish causes global and pericardial edema, and vascular defects characterized by reduced vascularization in intersegmental vessels and decreased sprouting in SIV vessels.

FA is currently used to prevent human cardiovascular malformations. The current study is an experimental one, with the main purpose of proving the cardiovascular toxicity of FA. In our experiments, extremely high doses of FA were used and compared to the currently used amounts for humans, and the side effects of overdosed FA were identified. Furthermore, we comparatively studied MTHF-Ca while analyzing FA. Unmetabolized FA in plasma has been found to occur regularly after supplementation with FA [46]. Our findings from the current study may raise concerns about excess FA supplementation. It is notable that the inhibitory effects of FA on cardiovascular development in the early embryonic stage was not found in MTHF-Ca. As MTHF-Ca is known to have no potentially adverse effects, this natural and biologically active form of folate seems more suitable for high-dose folate supplementation. The role of FA supplementation seems to be dual, with both beneficial and harmful effects depending upon its exposure duration and concentration. Therefore, the current study may provide useful information to indirectly explain why the previous studies have conflicting results.

## 5. Conclusions

The current study provides experimental evidence that FA, rather than MTHF-Ca, has cardiovascular toxicity in early embryonic development. This suggests that excessive supplementation of FA in perinatal women may be related to the risk of cardiovascular disorders in infants, such as CHD.

## Figures and Tables

**Figure 1 cells-11-03946-f001:**
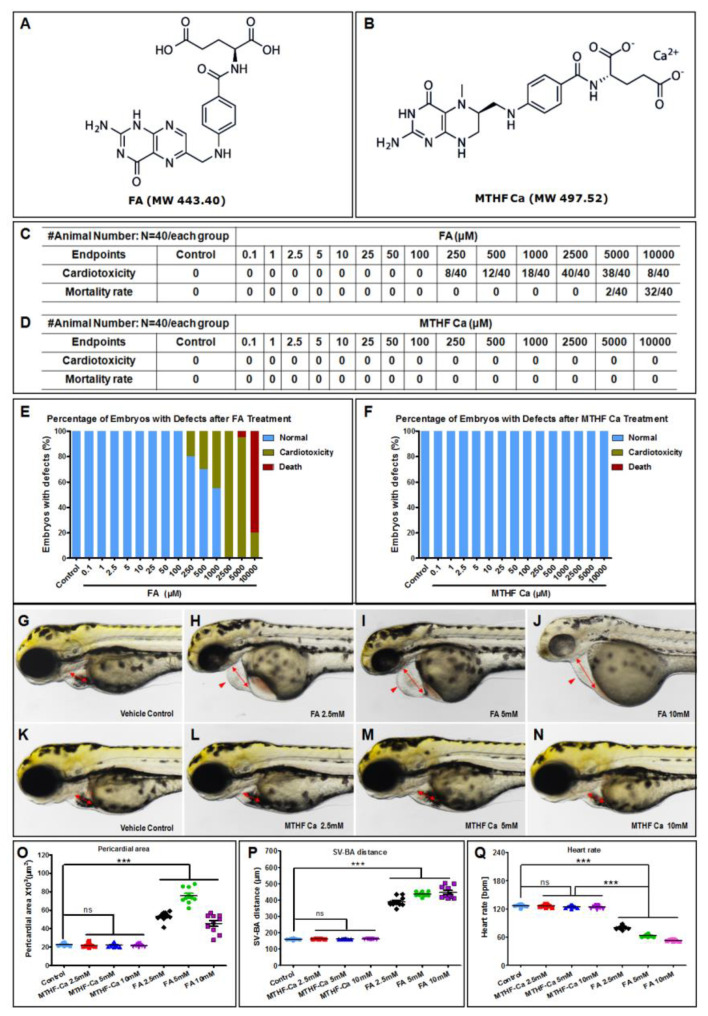
FA, but not MTHF-Ca, causes cardiotoxicity in vivo. (**A**) Chemical structural formulae of FA and (**B**) MTHF-Ca. FA, but not MTHF-Ca, caused cardiotoxicity in zebrafish at 3 dpf. (**C**,**E**) The endpoints of cardiotoxicity in FA were from 250 to 10,000 μM. The lowest cardiotoxicity at 250 μM of FA was 8/40, while the highest one was 40/40 at 2500 μM of FA. The mortality rates at 5000 and 10,000 μM of FA were 2/40 and 32/40. (**D**,**F**) There was no cardiotoxicity or embryonic defects in MTHF-Ca. (**G**–**N**) Representative bright-field images of zebrafish embryos at 3 dpf treated with vehicle control (fish water), FA (2.5 mM, 5 mM, and 10 mM) or MTHF-Ca (2.5 mM, 5 mM, and 10 mM). Compared with control or MTHF-Ca, FA-treated embryos presented pericardial oedema (red arrowheads), reduced contractile force, and reduced precardial blood congestion as sign of manifest heart failure. (Appendix A). Heartbeat and circulation in caudal vein (CV) was visible in the control fish and MTHF-Ca-treated fish but was abnormal in FA-treated fish. (**O**,**P**) Pericardial area and SV-BA distance in FA were increased compared to that in MTHF-Ca, (**Q**) while the heart rate was decreased. FA: folic acid; MTHF-Ca: 6S-5-methyltetrahydrofolate-calcium; SV-BA: sinus venosus and bulbus arteriosus; dpf, days postfertilization. *** *p* < 0.0001; ns: not significant.

**Figure 2 cells-11-03946-f002:**
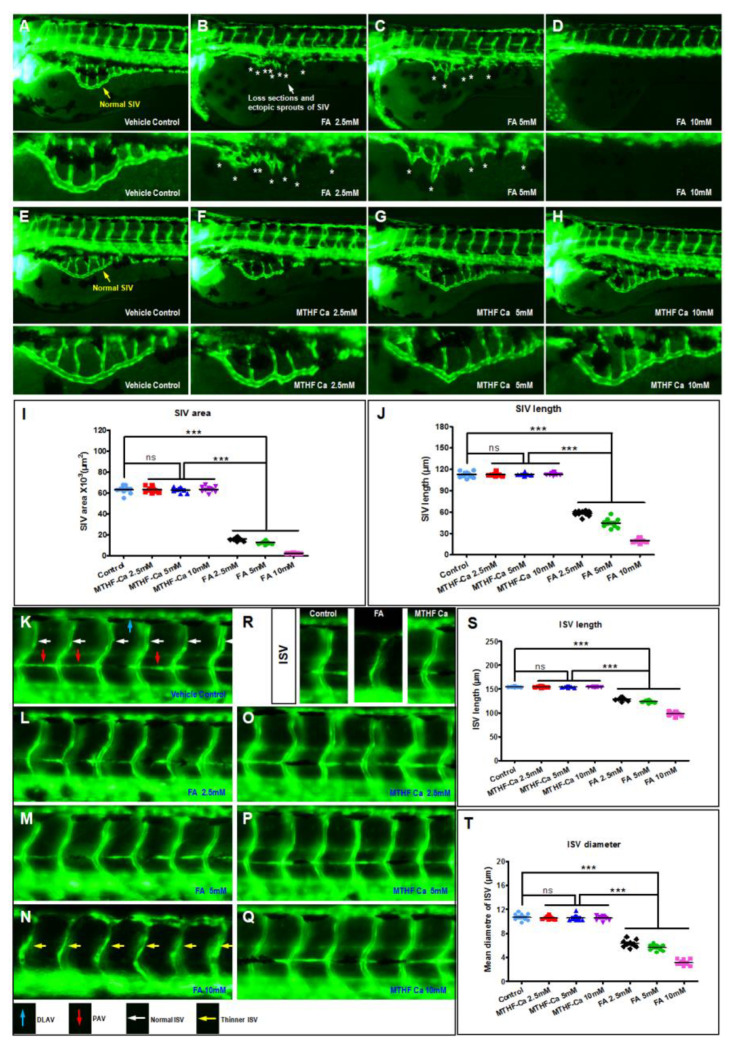
FA, but not MTHF-Ca, regulates vascular development in vivo. (**A**–**H**) FA, but not MTHF-Ca, impaired formation of zebrafish SIVs in a dose-dependent manner. ((**A**,**E**), yellow arrow) In control embryos, SIVs developed as a smooth, basket-like structure over the yolk at 3 dpf. ((**B**,**C**), asterisks) The embryos treated with FA presented a decreased number of ectopic SIV segments. (**I**,**J**) Quantification of the area and length of SIVs showed a significant decreased in FA-treated embryos. (**K**–**R**) The images of trunk regions taken at 3 dpf, with the vascular structures visualized by eGFP fluorescence and labeled ISV and DLAV, showed regular development in the embryo injected with control. (**S**,**T**) Compared with control or MTHF-Ca-treated groups, FA-treated embryos presented a lower number of incomplete and thinner ISVs ((**N**), yellow arrows). In control embryos, PAVs formed normally ((**K**), red arrows). Compared with control or MTHF-Ca, FA prevented PAV formation, the precursor to the lymphatic system. FA: folic acid; MTHF-Ca: 6S-5-methyltetrahydrofolate-calcium; DLAV: dorsal longitudinal anastomotic vessel; ISV: intersegmental vessel; PAV: parachordal vessels; SIV: subintestinal vein; dpf: days postfertilization. ns: not significant; *** *p* < 0.0001; ns: not significant.

**Figure 3 cells-11-03946-f003:**
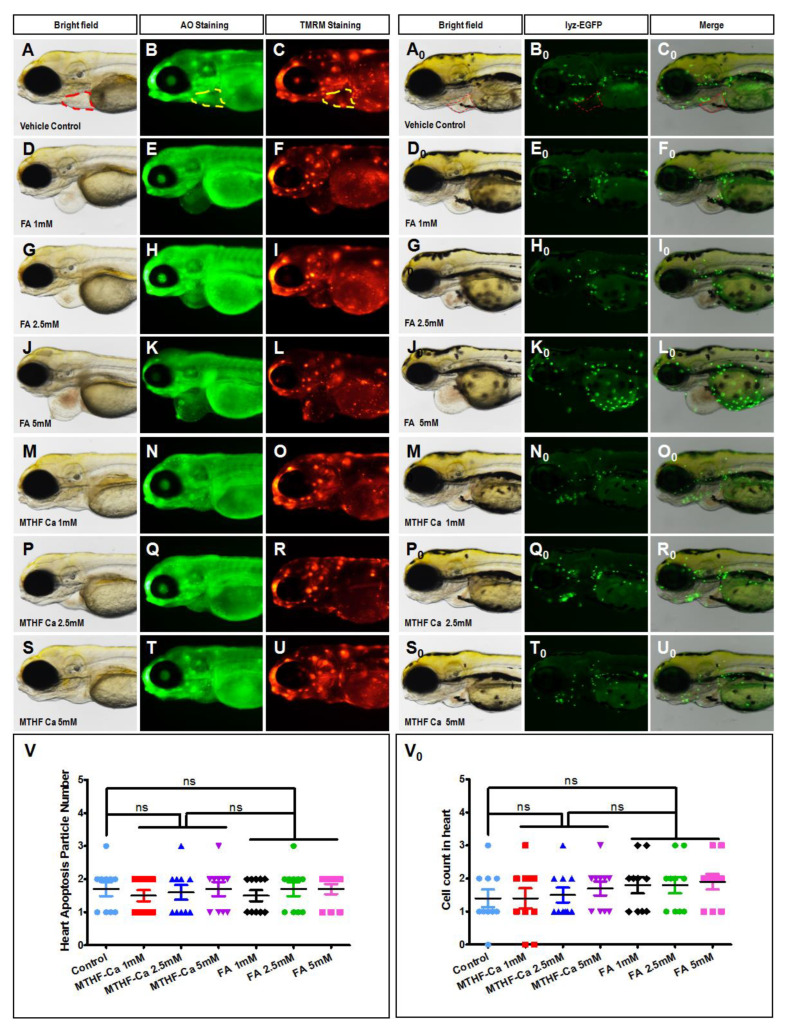
FA and MTHF-Ca do not induce heart-specific apoptosis and macrophage migration. (**A**–**U**) Representative bright-field and fluorescent images at 4 dpf treated with vehicle control (fish water), FA (1 mM, 2.5 mM, and 5 mM), or MTHF-Ca (1 mM, 2.5 mM, and 5 mM). Vehicle-control embryos and FA- or MTHF-Ca-treated embryos were stained with acridine orange (AO) at 4 dpf. The mitochondrial membrane potential (ΔΨm) was estimated by monitoring fluorescence aggregates of TMRM (tetramethyl rhodamine methyl ester, perchlorate, and biotium). FA and MTHF-Ca do not induce heart-specific mitochondrial defects. Heart apoptotic cells are visible as bright green spots, and less-bright homogenous green staining is unspecific background staining. Both FA- and MTHF-Ca-treated embryos exhibited few or no apoptotic cells in the heart. (Appendix A). (**V**) Quantification of apoptosis particle number in the heart. Error bars: SEM; (*n* = 10; ANOVA; ns: not significant). (**A**–**L**) Lateral view, anterior, left. (**A_0_**–**U_0_**) Representative bright-field and fluorescent images of *TG (zlyz:EGFP)* larvae at 4 dpf (dpf) treated with vehicle control (fish water), FA (1 mM, 2.5 mM, and 5 mM), or MTHF-Ca (1 mM, 2.5 mM, and 5 mM). Control fish showed the normal distribution of labeled cells ((**A_0_**–**C_0_**), circle area). Compared with vehicle-control fish, FA- or MTHF-Ca-treated fish also showed the normal distribution of labeled cells. (Appendix A). (**V_0_**) Quantification of the macrophage number in the heart. Columns: means; bars: SEM (*n* = 10; ANOVA). FA: folic acid; MTHF-Ca: 6S-5-methyltetrahydrofolate-calcium. ns: not significant.

**Figure 4 cells-11-03946-f004:**
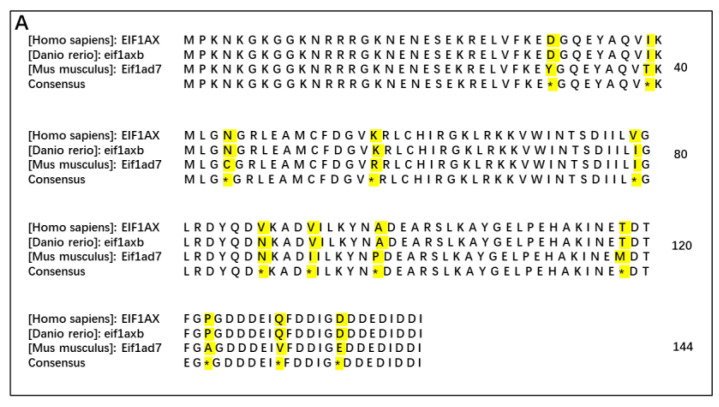
FA, but not MTHF-Ca, downregulates *eif1axb* expression in a time-specific manner. (**A**) There is a high homology among humans (*EIF1AX*), zebrafish (*eif1axb*), and mice (*Eif1ad7*). Amino acid sequences of *eif1axb* orthologs from three species were aligned. The un-conserved amino acids are highlighted in yellow. (**B**) This finding was confirmed by RT-PCR amplification in zebrafish. The expression of *eif1axb* in zebrafish was significantly downregulated by FA treatment at three embryo development stages (12 hpf, 24 hpf, and 48 hpf) (*n* = 30 individual embryos). FA: folic acid; MTHF-Ca: 6S-5-methyltetrahydrofolate-calcium; L: low dose; H: high dose. *** *p* < 0.001; ** *p* < 0.01; ns: not significant; hpf: hours postfertilization.

**Figure 5 cells-11-03946-f005:**
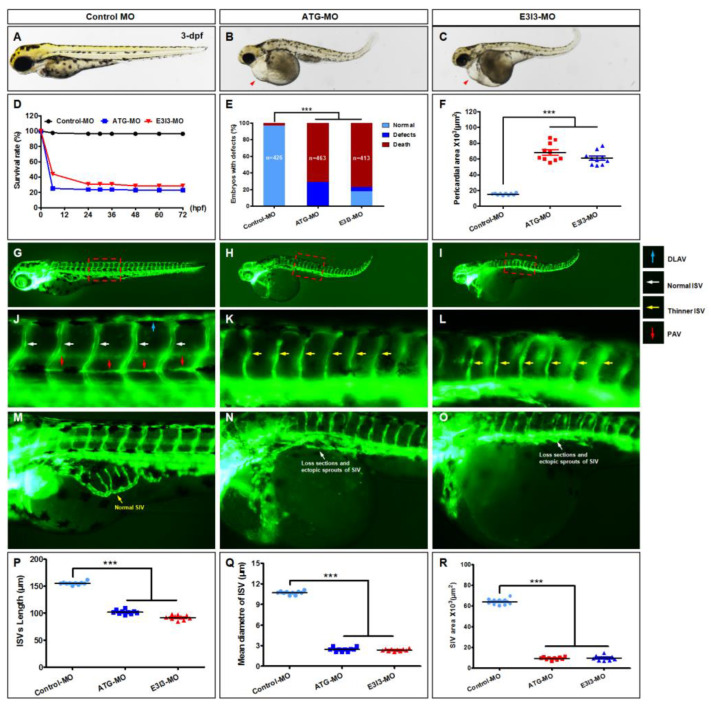
Loss of *eif1axb* phenocopies FA-induced cardiovascular defects. (**A**–**C**) Gross morphology of zebrafish embryos at 3 dpf. Compared with control zebrafish, *eif1axb* knockdown causes pericardial oedema ((**B**,**C**), red arrowheads), reduced contractile force, and reduced precardial blood congestion as signs of manifest heart failure. (**D**) A time–course plot of percent survival in control vs. *eif1axb* morphants for 3 days. Panel (**E**) shows the percentage of embryos with developmental defects. (**F**) Quantification of the pericardial area of embryos. Error bars: means ± SEM. *** *p* < 0.001 (*n* = 10; ANOVA.). (**G**–**O**) Representative fluorescent images of *Tg(fli1a:EGFP)^y1^* embryos at 3 dpf. (**G**,**J**) Images of trunk regions taken at 3 dpf, with the vascular structures visualized by eGFP fluorescence and labeled ISV and DLAV, showed regular development in the embryo injected with control MO. Compared with control MO, embryos injected with *eif1axb*-MO presented thinner ISVs ((**K**,**L**), yellow arrows). In control embryos, the parachordal vessels (PAVs) formed normally ((**J**), red arrows). Compared with control, MO knockdown *eif1axb* prevented PAV formation, the precursor to the lymphatic system (**K**,**L**). The boxed regions are shown at higher magnification in the bottom panels. In control embryos, SIV vessels developed as a smooth, basket-like structure over the yolk at 3 dpf ((**A**,**E**), yellow arrow). In contrast, embryos injected with *eif1axb*-MO presented a decreased number of ectopic SIV segments ((**M**), asterisks). (**P**,**Q**) Quantification of the mean length and diameter of ISVs showed a significant decrease in *eif1axb* morphants. Columns: mean ± SEM (*n* = 10; ANOVA; *** *p* < 0.001). (**R**) Quantification of the area of SIVs showed a significant decrease in *eif1axb* morphants. Columns: mean; bars: SEM (*n* = 10; ANOVA; *** *p* < 0.001). DLAV: dorsal longitudinal anastomotic vessels; ISV: intersegmental vessel; dpf: days postfertilization.

**Figure 6 cells-11-03946-f006:**
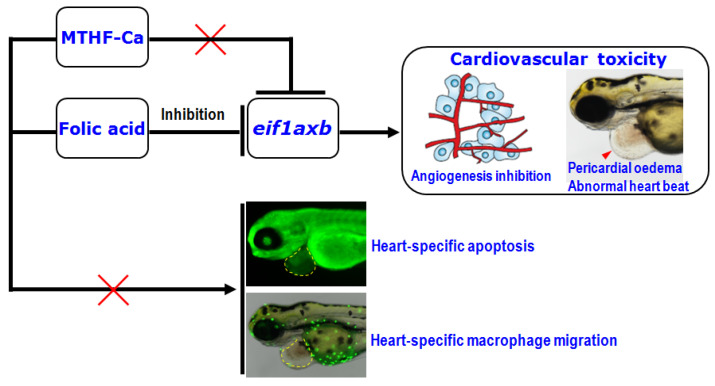
A putative molecular mechanism of cardiovascular toxicity in excessive supplementation of folic acid. This is a schematic illustration to demonstrate that excessive FA treatment has cardiovascular toxicity and lead to angiogenesis inhibition in zebrafish, but not apoptosis and macrophage migration in the heart. Downregulation of an essential eukaryotic translation initiation factor (*eif1axb* in zebrafish) by FA may explain the molecular mechanism of cardiovascular toxicity. No adverse effects of MTHF-Ca is seen.

## Data Availability

All the data in this study are available in the figures and Appendix A of this paper.

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
