# Peer review of "Evaluation of Cardiovascular Toxicity of Folic Acid and 6S-5-Methyltetrahydrofolate-Calcium in Early Embryonic Development"

_cells, 2022, doi:10.3390/cells11243946_

Round 1

Reviewer 1 Report

In this manuscript by Cheng et al, the authors examined the effects of folic acid (FA) and Ca-methyltetrahydrofolate (MTHF-Ca) on heart development in zebrafish and mouse development models. The authors conclude that excessive FA intake may be an unrecognized risk factor for defects in heart development. This is an interesting paper, with potentially valuable findings, that is severely hampered by the need for English language proofreading. It is difficult in some places to evaluate the methodology and results due to confusing language.

Specific concerns:

Line 69-72: The end of this statement is incorrect. FA requires reduction to tetrahydrofolate to be used, not MTHF. THF can be used in many aspects of folate metabolism without conversion to MTHF. Also, DHFR should be introduced here as the enzyme that reduces FA, as it is discussed later in the paper (e.g. line 94), without any explanation of its significance.

Line 108-116 (Methods Section 2.1): What is the source of the zebrafish? What are they fed?

Line 112-115 about mice does not belong in this section (should be in 2.5).

Line 119-124: It is stated in Line 259-264 that 20 mM NaHCO3 is used as a solvent for FA and MTHF. This should be included in the methods section, and it should be made clear that the control group is a solvent control group. If it is not a solvent control group in any of the experiments, it should be stated.

Line 119-121: How many zebrafish were used per concentration/timepoint?

Line 156: TMRM staining for mitochondrial membrane potential is described in the methods, and mentioned in Fig 3, but the results are not presented or described anywhere in the paper.

Line 166-182 (Methods Section 2.5): A more complete description of the diet of the mice is required. Is it a natural ingredient diet, casein-based, or amino acid defined? What is folate content, and is it in the form of folic acid, naturally occurring folates, or a mix? The background folate content of the diet may affect the interpretation of results.

Line 173-175: The FA and MTHF solutions are not equimolar – why not? The authors are aware that the molecular weights of the 2 compounds are different, as they supply them at line 259. The supplemental dose of MTHF is only 89% of the FA dose, which could lead to a false negative result for the high dose mice. This should be addressed in the discussion. Also, was 20 mM NaHCO3 or distilled water used as a solvent for FA and MTHF in this experiment? The use of distilled water as the control group implies that water is the solvent, but this should be clear. Finally, how were the solutions administered to the mice? Gavage?

Line 175: 1 week is not enough time to equilibrate tissue folates to the supplemental dose. This can take 4-5 weeks. The authors must acknowledge somewhere that the concentration of folates in many tissues would be constantly changing over the course of gestation. This is especially important because, based on Figure 4A, some of the litters were collected as far as 2 weeks after most of the others were born, and therefore exposed to different maternal conditions.

Line 177: It appears that the males were not separated from the females after the vaginal plug was observed. Why not? This will confound the evaluation of the length of gestation as in Figure 4A.

Line 178-179: “After female mice had given birth anesthesia with pentobarbital sodium (60 mg/kg), while heart rates were calculated with the electrocardiogram…” as written, it is not clear if they mean the mothers or the pups. Also, how were the pups euthanized?

Line 184-186: A reference for the superovulation and fertilization procedures should be included.

Line 187: Again, the FA and MTHF solutions are not equimolar, so the blastocysts in the MTHF group are exposed to a lower concentration than the FA group.

Line 225-231 (Methods Section 2.8): qRT-PCR requires normalization to one or preferably more reference genes (Bustin et al, Clinical Chemistry, 2009). What genes were used as normalizers and how where they chosen?

Line 234: Total RNA from what?

Line 270-273: “…while exposure at the highest concentration (2500 μM of FA) had a 100% lethal response frequency…” as written is confusing, and does not accurately describe the data shown in Figure 1C.  The table in Figure 1C shows 100% cardiotoxicity for 2500 μM FA, as well as mortality of 2/40 and 32/40 for 5000 and 10000 μM FA respectively, and 100% cardiotoxicity in the surviving embryos. The authors underplay their MTHF results here and in the legend of Figure 1, as they neglect to point out that there is also no mortality among embryos at the higher MTHF concentrations.

Figure 1B: The CH3 group of MTHF is missing.

Line 365-368 and Figure 1A: The methods state at line 176-177 that vaginal plugs were checked to establish E0.5, so why are these results presented as days after mating rather than days post coitum? Days post coitum would be more useful to determine if the treatment altered the length of gestation, and if plugged females failed to establish a pregnancy (% of plugged mice that gave birth by 21 days after gestation). However, since the males were apparently not separated from the females at E0.5, it is impossible to be sure that the pregnancy dates from the observed plug, and not a subsequent copulation.

The statement “…while the other females got pregnant for a relatively long time” does not make sense in English. The longer length of time between mating and birth for some mice would reflect both variation in the stage of the estrus cycle of the female at mating, and possibly a second mating after the failure of the prior copulation or pregnancy. It is not clear what the authors are suggesting with this observation of delayed litters.

Line 368-369: C57Bl/6 mice are notoriously matricidal, particularly if stressed, or if the pups have defects. Were any steps taken to account for matricide when determining litter size? When were the litter sizes evaluated?

Line 370-373: Heart defect incidence has been reported to vary between males and females. Was the sex of the pups considered when selecting them for ECG and other evaluations?

Line 374-376: How many hearts were evaluated by histopathology? How many heart defects were observed (as written, suggests only 1)?

Line 403-404: It would be useful to state here that Berry and Ayling reported rat liver DHFR activity of 577 nmol THF/min/g wet weight liver with DHF as a substrate, which was 35x higher that human activity of 16.3 nmol THF/min/g wet weight liver. In contrast, the authors report DHFR activities ranging from 70 to 220 nmol THF/min/g wet weight liver. It would be useful if they would report the average value for comparison. This is valuable information that suggests that mouse DHFR activity is closer to human than rat.

Line 422 and 442: The wording is confusing. Do the authors mean that eif1axb knockdown phenocopies FA-induced cardiovascular defects?

Line 493-495: This explanation for the FA dose administered should be described in more detail and included in the methods section. What are the human doses that are modeled here? Is the folate content of the mouse diet factored into this calculation?

Line 498-499: This detail should be included in the methods for the in vitro culture. However, what is the relevance of human cord blood folate levels to pre-implantation mouse embryos?

Line 508-512: It is not clear what the authors are discussing here. Do they mean that in mice specifically, decreased eif1ad7 expression leads to lethality rather than specific malformations? How does that reconcile with the specific malformations observed in zebrafish? This could be evaluated by examining mouse embryos at earlier time points for defects and viability. It is possible that the early lethality proposed here is due to severe cardiac malformations leading to death in utero.

All Figures: The panels showing graphs are too small to evaluate properly, and the font size is too small to read in many panels, even when zoomed in.

Legends of Figure 1, 2 : n and statistics details that are shown in the other figures are missing for these figures. The n are also missing from the legend of Figure 4 for panel B.

Supplemental Figure S4: the legend does not provide descriptions of the panels.

Supplemental Figure S5: Panels A and B and parts of the legend are copy-pasted directly from the original Bailey and Ayling paper. This should not be done, and there is no need for it, as the original paper is open access. The authors should limit themselves to summarizing the previous results and referencing the paper.

Supplemental Figure S6: The legend does not describe what is shown in panel A.

Minor concerns:

Line 72: Should say Evidence from, not “Evidence form…”

Line 73-4: This sentence makes no sense

Line 92-94: The analysis of differentially expressed genes is an interesting experiment, and fits well in the manuscript; however, the sentence “We analyzed differentially expressed genes… because the activity of DHFR in rodents is high” as written does not make sense as a rationale for this experiment. The rationale is described again at line 404-406, but the author’s thinking that links DHFR activity and the need to analyze differentially expressed genes is still not clear.

Line 146: should be “effect”, not “efficacy”?

Line 277: This should say that representative images are shown in Figure S1

Line 353: “botch” should be “both”. Also, “In contrast” to what? There are no differences between any of the groups.

Line 289: “Pencardial” should be “Pericardial”?

Line 419: eif1axb should be italicized.

Author Response

Dear Reviewer,

Thank you very much for your valuable comments and constructive suggestions on our manuscript “Evaluation of Cardiovascular Toxicity of Folic Acid and 6S-5-Methyltetrahydrofolate-Calcium in Early Embryonic Development”. (MDPI Cell-2013795). The manuscript has been revised as suggested by the reviewers and please find our responses point by point as below. We hope that this revised manuscript has been improved for publication in your journal.

Best regards!

Sincerely yours,

Harvest F. Gu

In this manuscript by Cheng et al, the authors examined the effects of folic acid (FA) and Ca-methyltetrahydrofolate (MTHF-Ca) on heart development in zebrafish and mouse development models. The authors conclude that excessive FA intake may be an unrecognized risk factor for defects in heart development. This is an interesting paper, with potentially valuable findings, that is severely hampered by the need for English language proofreading. It is difficult in some places to evaluate the methodology and results due to confusing language.

Specific concerns:

Line 69-72: The end of this statement is incorrect. FA requires reduction to tetrahydrofolate to be used, not MTHF. THF can be used in many aspects of folate metabolism without conversion to MTHF. Also, DHFR should be introduced here as the enzyme that reduces FA, as it is discussed later in the paper (e.g. line 94), without any explanation of its significance.

Thank, the statement has been removed. DHFR is now introduced in the introduction and the reference [8] has been moved ahead.

Line 108-116 (Methods Section 2.1): What is the source of the zebrafish? What are they fed?

Zebrafish is from SMOC and fed with artemia. This has been added.

Line 112-115 about mice does not belong in this section (should be in 2.5).

Mice should not belong to this section. Sorry for our mistake.

Line 119-124: It is stated in Line 259-264 that 20 mM NaHCO3 is used as a solvent for FA and MTHF. This should be included in the methods section, and it should be made clear that the control group is a solvent control group. If it is not a solvent control group in any of the experiments, it should be stated.

We have used NaHCO3 as a solvent for FA for stock solution. The working solution of FA, however, is prepared with water. In mice experiments, FA and MTHF were ig given with water. 

Line 119-121: How many zebrafish were used per concentration/timepoint?

N=40 each group.

Line 156: TMRM staining for mitochondrial membrane potential is described in the methods, and mentioned in Fig 3, but the results are not presented or described anywhere in the paper.

FA or MTHF-Ca doesn’t induce heart-specific apoptosis and macrophage migration in zebrafish as seen in Suppl Info.

Line 166-182 (Methods Section 2.5): A more complete description of the diet of the mice is required. Is it a natural ingredient diet, casein-based, or amino acid defined? What is folate content, and is it in the form of folic acid, naturally occurring folates, or a mix? The background folate content of the diet may affect the interpretation of results.

The diet is SPF class and contains 5.8 mg/kg FA. More detailed information has been added.

Line 173-175: The FA and MTHF solutions are not equimolar – why not? The authors are aware that the molecular weights of the 2 compounds are different, as they supply them at line 259. The supplemental dose of MTHF is only 89% of the FA dose, which could lead to a false negative result for the high dose mice. This should be addressed in the discussion. Also, was 20 mM NaHCO3 or distilled water used as a solvent for FA and MTHF in this experiment? The use of distilled water as the control group implies that water is the solvent, but this should be clear. Finally, how were the solutions administered to the mice? Gavage?

Sorry, it was error. We have corrected the calculations. In mice experiments, FA is intragastric administration with distilled water. In control group, the mice are in distilled water ig,

Line 175: 1 week is not enough time to equilibrate tissue folates to the supplemental dose. This can take 4-5 weeks. The authors must acknowledge somewhere that the concentration of folates in many tissues would be constantly changing over the course of gestation. This is especially important because, based on Figure 4A, some of the litters were collected as far as 2 weeks after most of the others were born, and therefore exposed to different maternal conditions.

Line 177: It appears that the males were not separated from the females after the vaginal plug was observed. Why not? This will confound the evaluation of the length of gestation as in Figure 4A.

The original Figure 4A has been removed.

Line 178-179: “After female mice had given birth anesthesia with pentobarbital sodium (60 mg/kg), while heart rates were calculated with the electrocardiogram…” as written, it is not clear if they mean the mothers or the pups. Also, how were the pups euthanized?

We have added the relevant information in line 174-176.

Line 184-186: A reference for the superovulation and fertilization procedures should be included.

Yes, a reference [31] has been added.

Line 187: Again, the FA and MTHF solutions are not equimolar, so the blastocysts in the MTHF group are exposed to a lower concentration than the FA group.

In the blastocyte culture experiments, the solutions of FA and MTHF are equimolar.

Line 225-231 (Methods Section 2.8): qRT-PCR requires normalization to one or preferably more reference genes (Bustin et al, Clinical Chemistry, 2009). What genes were used as normalizers and how where they chosen?

Line 234: Total RNA from what?

The reference gene name and tissues for RNA extraction have been added in line 208 and 225.

Line 270-273: “…while exposure at the highest concentration (2500 μM of FA) had a 100% lethal response frequency…” as written is confusing and does not accurately describe the data shown in Figure 1C.  The table in Figure 1C shows 100% cardiotoxicity for 2500 μM FA, as well as mortality of 2/40 and 32/40 for 5000 and 10000 μM FA respectively, and 100% cardiotoxicity in the surviving embryos. The authors underplay their MTHF results here and in the legend of Figure 1, as they neglect to point out that there is also no mortality among embryos at the higher MTHF concentrations.

Sorry for the confusing and we have modified the description.

Figure 1B: The CH3 group of MTHF is missing.

We have checked it out. Figure 1B is correct according to the generals of Chemistry.

Line 365-368 and Figure 1A: The methods state at line 176-177 that vaginal plugs were checked to establish E0.5, so why are these results presented as days after mating rather than days post coitum? Days post coitum would be more useful to determine if the treatment altered the length of gestation, and if plugged females failed to establish a pregnancy (% of plugged mice that gave birth by 21 days after gestation). However, since the males were apparently not separated from the females at E0.5, it is impossible to be sure that the pregnancy dates from the observed plug, and not a subsequent copulation.

The statement “…while the other females got pregnant for a relatively long time” does not make sense in English. The longer length of time between mating and birth for some mice would reflect both variation in the stage of the estrus cycle of the female at mating, and possibly a second mating after the failure of the prior copulation or pregnancy. It is not clear what the authors are suggesting with this observation of delayed litters.

The original Figure 4A has been removed, and the text has been changed.

Line 368-369: C57Bl/6 mice are notoriously matricidal, particularly if stressed, or if the pups have defects. Were any steps taken to account for matricide when determining litter size? When were the litter sizes evaluated?

Line 370-373: Heart defect incidence has been reported to vary between males and females. Was the sex of the pups considered when selecting them for ECG and other evaluations?

The litter size is evaluated based upon the observation and anatomy examination under a microscope with Moticam5+. In addition, we bought the instruments such as RS-NM ECF (iworx), IX-BI04 (iworx), MRBP (IIPC) from USA for the experiments with mice.

We did not separate male and female of the pups. This point has been taken into our consideration in further experiments. Thank!

Line 374-376: How many hearts were evaluated by histopathology? How many heart defects were observed (as written, suggests only 1)?

N=3, each group.

Line 403-404: It would be useful to state here that Berry and Ayling reported rat liver DHFR activity of 577 nmol THF/min/g wet weight liver with DHF as a substrate, which was 35x higher that human activity of 16.3 nmol THF/min/g wet weight liver. In contrast, the authors report DHFR activities ranging from 70 to 220 nmol THF/min/g wet weight liver. It would be useful if they would report the average value for comparison. This is valuable information that suggests that mouse DHFR activity is closer to human than rat.

Yes, we fully agree with you. The average value is 130.21±22.97. This information has been added in supple info.

Line 422 and 442: The wording is confusing. Do the authors mean that eif1axb knockdown phenocopies FA-induced cardiovascular defects?

Sorry, and we have made a correction.

Line 493-495: This explanation for the FA dose administered should be described in more detail and included in the methods section. What are the human doses that are modeled here? Is the folate content of the mouse diet factored into this calculation?

Yes, we calculated the amount for FA administration according to human doses and BSA of mouse (K=9.1).

Line 498-499: This detail should be included in the methods for the in vitro culture. However, what is the relevance of human cord blood folate levels to pre-implantation mouse embryos?

In our experiments, the data may not explain the relevance of human cold blood folate. The reference and related discussion have been removed.

Line 508-512: It is not clear what the authors are discussing here. Do they mean that in mice specifically, decreased eif1ad7 expression leads to lethality rather than specific malformations? How does that reconcile with the specific malformations observed in zebrafish? This could be evaluated by examining mouse embryos at earlier time points for defects and viability. It is possible that the early lethality proposed here is due to severe cardiac malformations leading to death in utero.

All Figures: The panels showing graphs are too small to evaluate properly, and the font size is too small to read in many panels, even when zoomed in.

We have added all figures at tiff format in a large size. They should be clear.

Legends of Figure 1, 2 : n and statistics details that are shown in the other figures are missing for these figures. The n are also missing from the legend of Figure 4 for panel B.

Supplemental Figure S4: the legend does not provide descriptions of the panels.

Supplemental Figure S6: The legend does not describe what is shown in panel A.

All figures in suppl info have been checked and corrected.

Supplemental Figure S5: Panels A and B and parts of the legend are copy-pasted directly from the original Bailey and Ayling paper. This should not be done, and there is no need for it, as the original paper is open access. The authors should limit themselves to summarizing the previous results and referencing the paper.

We have removed the copied figures from Bailey and Avling paper. Thank!

Minor concerns:

Line 72: Should say Evidence from, not “Evidence form…”

Line 73-4: This sentence makes no sense

Line 92-94: The analysis of differentially expressed genes is an interesting experiment, and fits well in the manuscript; however, the sentence “We analyzed differentially expressed genes… because the activity of DHFR in rodents is high” as written does not make sense as a rationale for this experiment. The rationale is described again at line 404-406, but the author’s thinking that links DHFR activity and the need to analyze differentially expressed genes is still not clear.

Line 146: should be “effect”, not “efficacy”?

Line 277: This should say that representative images are shown in Figure S1

Line 353: “botch” should be “both”. Also, “In contrast” to what? There are no differences between any of the groups.

Line 289: “Pencardial” should be “Pericardial”?

Line 419: eif1axb should be italicized.

Thank you so much for all these corrections! They are small but we have learnt from you because you did the reviewing very detailed and very carefully.

Reviewer 2 Report

This is an extremely important study examining the effects of FA and 6S-5-MTHF on the development of heart malformations and cardiac function in early pregnancy. It is well known that folic acid has a preventive effect on neural tube defects. Next, many clinical trials have been conducted to investigate the preventive effect of folic acid on the development of hear malformations. Now there is growing hope that the serious cardiac malformations can be prevented. This study shows that there is a difference in the biological functions of these two substances. In this sense, I believe this is an important study.

I would like to point out the followings:

1) Currently, the human dosage of synthetic folic acid should not exceed 1000 µg. a day  Any higher dose over 1000 μg may cause severe biological effects. The activities of enzymes related to one carbon metabolism are naturally different in zebrafish, mice, and humans. However, the high dose of folic acid and MTHF in this experiment should be considered pharmacological doses. Therefore, the concentrations of these substances used in the zebrafish and mouse experiments should be shown to correspond to that in humans. In reality, the amount of these substances would never be used or administered to humans

 2) The charts and figures are too small to distinguish clearly. It should be made large enough for us to examine.

Author Response

To Reviewer 2

This is an extremely important study examining the effects of FA and 6S-5-MTHF on the development of heart malformations and cardiac function in early pregnancy. It is well known that folic acid has a preventive effect on neural tube defects. Next, many clinical trials have been conducted to investigate the preventive effect of folic acid on the development of hear malformations. Now there is growing hope that the serious cardiac malformations can be prevented. This study shows that there is a difference in the biological functions of these two substances. In this sense, I believe this is an important study.

Thank you very much! We hope that this study concerning cardiovascular toxicity of folic acid in early embryonic development will attract the attention of researchers and government officials. Excessive supplementation of folic acid in perinatal women may be related to the potential risk of cardiovascular disorders such as congenital heart disease.

I would like to point out the followings:

  • Currently, the human dosage of synthetic folic acid should not exceed 1000 µg. a day Any higher dose over 1000 μg may cause severe biological effects. The activities of enzymes related to one carbon metabolism are naturally different in zebrafish, mice, and humans. However, the high dose of folic acid and MTHF in this experiment should be considered pharmacological doses. Therefore, the concentrations of these substances used in the zebrafish and mouse experiments should be shown to correspond to that in humans. In reality, the amount of these substances would never be used or administered to humans.

Thank you very much! This comment is very important. In humans, rats, mice and zebrafish, the activity of dihydroreductase is very different. Among the same species, the individual difference is very large. The current study is an experimental one, with the main purpose of proving the cardiovascular toxicity of folic acid. In the future clinical application study, we will certainly adopt your suggestion.

  • The charts and figures are too small to distinguish clearly. It should be made large enough for us to examine.

Sorry! We have now attached all figures at tiff format and with large size. They should look clear.

Round 2

Reviewer 1 Report

This manuscript has been improved from the previous version, but there is still missing information about the in vivo mouse study. The zebrafish study is very interesting, and clearly shows the different effects of folic acid and MTHFR-Ca.

Line 168-169: The description of the mouse diet is lacking details to allow comparison to other studies. What type of diet is it (e.g.: natural ingredient diet, casein-based, or amino acid defined)? The acronym SPF should be defined. It often refers to specific pathogen free, which does not give any indication of the nutritional content of the diet.

Line 169 and 499: The authors added the detail at line 169 that the diet contains 5.8 mg/kg folic acid. This is quite a high amount - almost 3x the amount recommended for AIN-93 (2 mg/kg), which is the amount used in the control diets of many of the mouse studies cited in this paper. Estimating an average intake of 3 g/d for a 20 g female, this diet would result in a folic acid intake of about 17.4 ug/d, as compared to 6 ug/day for AIN-93. In contrast, the low and high dose FA supplements would be about 1.2 ug/day and 2.4 ug/day. At line 499 they state that the high dose FA mice get double the low dose, but this is not actually correct. The supplemented amount is double, but the actual folic acid intake of the high dose mice is only approximately 6% higher than the low dose due to the FA content of the mouse diet. It is not clear from the methods as written that the authors factored the diet content into their calculations, and this may have affected outcomes. This should be addressed in the discussion.

Line 173: Giving the supplement by gavage is a strength to this study, because it more accurately mimics how people use vitamin supplements than including it in the food.

Line 171-172 and 185-186: The authors correction at line 171-172 clarifies that the FA and MTHF-Ca supplements were given to the mice in equimolar amounts but there was no correction at line 185-186. They state in their response that “In the blastocyte culture experiments, the solutions of FA and MTHF are equimolar.” However, the paper at line 185-186 states: “The fertilized eggs were randomly grouped for control with potassium simplex optimized medium (KSOM) medium (Elite-Media), KSOM-FA (60 ng/L) and KSOM-MTHF-Ca (60 ng/L).” Either this sentence requires correction, or the concentrations are not equimolar.

Line 407-408 (Figure 4): It would be helpful to point out the heart defect, as was done in the zebrafish pictures.

Line 497-499: The authors state that “the low dose of FA administered to mice was calculated according to a clinically relevant human dose based upon the body surface area, while high dose of FA is double.” What is the human dose that they are modelling? I understand from the response that they did a BSA conversion calculation, but they should state what the human dose was in the paper.

Author Response

Dear Reviewer,

Thank you very much for your advanced comments! We have finally decided to remove the information about the in vivo mouse study. The data of this paper may become more clear. In addition we have stated that this is an experimental study in discussion (line 481-484). We hope this revised manuscript can be now accepted for publication in your journal.

Best regards!

All authors

Reviewer 2 Report

Thank you of your revised article and answer.

Folic acid is currently used to prevent human cardiovascular malformations. In this experiment, extremely higher doses of folic acid and MTHF were used compared to currently used amount for human,  and  the side effects of overdosed folic acid have been identified.
Therefore, the text (*) presented in your response to my comment should be included in the discussion.

(* The current study is an experimental one, with the main purpose of proving the cardiovascular toxicity of folic acid. In the future clinical application study, we will certainly adopt your suggestion.)

The enlarged charts figures are fine to be read.

Author Response

Dear Reviewer,

We agree with you and have added the text in discussion (line 481-484).

Thank you very much and best regards!

All authors